# Temperature Effects on Optical Properties and Chemical Composition of Secondary Organic Aerosol Derived from *n*-Dodecane

Junling Li[1,2,4], Weigang Wang[*,1,2], Kun Li[5,a], Wenyu Zhang[1,2], Chao Peng[1,2], Li Zhou[6], Bo Shi[1,2], Yan Chen[1,2], Mingyuan Liu[1,2], Hong Li[4], Maofa Ge[1,2,3]

[1] State Key Laboratory for Structural Chemistry of Unstable and Stable Species, Beijing National Laboratory for Molecular Sciences (BNLMS), CAS Research/Education Center for Excellence in Molecular Sciences, Institute of Chemistry, Chinese Academy of Sciences, Beijing 100190, P. R. China
[2] University of Chinese Academy of Sciences, Beijing 100049, P. R. China
[3] Center for Excellence in Regional Atmos. Environ., Institute of Urban Environment, Chinese Academy of Sciences, Xiamen, 361021, P. R. China
[4] State Key Laboratory of Environmental Criteria and Risk Assessment, Chinese Research Academy of Environmental Sciences, Beijing 100012, P. R. China
[5] Air Quality Research Division, Environment and Climate Change Canada, Toronto, Ontario M3H5T4, Canada
[6] College of Architecture and Environment, Sichuan University, Chengdu, P. R. China
[a] now at: Laboratory of Atmospheric Chemistry, Paul Scherrer Institute (PSI), 5232 Villigen, Switzerland

*Correspondence to*: Weigang Wang (wangwg@iccas.ac.cn)

**Abstract.** Environmental temperature plays a vital role in controlling chemical transformations that lead to the formation of secondary organic aerosol (SOA), and ultimately impact composition and optical properties of the aerosol particles. In this study, we investigated optical properties of *n*-dodecane secondary organic aerosol under two temperature conditions: 5 ˚C and 25 ˚C. It was shown that low temperature could enhance the real part of refractive index (RI) of the SOA at the wavelengths of 532 nm and 375 nm. Mass spectrometry analysis revealed that molecular composition of *n*-dodecane SOA was significantly modified by temperature: a large amount of oligomers were formed under low temperature condition, which leaded to higher RI values. These findings will help improve our understanding of the chemical composition and optical properties of SOA under different temperature conditions, and provide one possible explanation of the low visibility in suburban areas during winter.

## 1 Introduction

Organic aerosol, especially secondary organic aerosol (SOA), plays a vital role in air quality, climate change, and human health (Kanakidou et al., 2005; Poschl, 2005; Mellouki et al., 2015; Poschl and Shiraiwa, 2015; von Schneidemesser et al., 2015; Shrivastava et al., 2017). SOA accounts for a high proportion of atmospheric particulate matter around the world, especially in the heavily polluted areas (Liu et al., 2017; Sun et al., 2014; Huang et al., 2014). Due to the variety of precursors and oxidation pathways, the composition of SOA is very complicated and variable (Lu, 2018; von Schneidemesser et al., 2015; Poschl and Shiraiwa, 2015; Hallquist et al., 2009; George et al., 2015), and the optical properties of SOA also exhibit different characteristics (Shrivastava et al., 2017; Zhang et al., 2015; Moise et al., 2015; Laskin et al., 2015). The complex refractive

index (RI), m= n + ki (n is the real part, and k is the imaginary part; they express the extent of scattering and absorbing, respectively), is the only intrinsic optical property of a particle. RI is controlled by the chemical composition and physical characteristics (e.g., morphology and shape) of a particle (Moise et al., 2015). Quantifying the RI of aerosol particles is highly important to evaluate their optical properties, and further estimate their impacts on atmospheric visibility and Earth's radiative balance.

Aerosol physicochemical properties are strongly dependent on the atmospheric conditions, e.g., relative humidity (Ervens et al., 2011; Sun et al., 2014), temperature (Wang et al., 2017), and oxidizing conditions (oxidant type, e.g., $NO_3$, OH, $O_3$; oxidation concentrations, e.g., photochemical age) (Cheng et al., 2016; Shrivastava et al., 2017; George et al., 2015; Kanakidou et al., 2005). Therefore, it is important to study the SOA formation and optical properties under varying atmospheric conditions to simulate the processes in the real atmosphere. There have been a number of smog chamber experiments on the effect of seed particles (Huang et al., 2017; Denjean et al., 2014; Song et al., 2013; Lee et al., 2013; Li et al., 2018; Li et al., 2017a; Trainic et al., 2011), oxidant type (e.g., $NO_3$ (Peng et al., 2018; Lu et al., 2011), OH (Liu et al., 2015; Lin et al., 2015; Li et al., 2014; Nakayama et al., 2013; Cappa et al., 2011), and $O_3$ (Peng et al., 2018; Kim et al., 2014; Flores et al., 2014; Kim and Paulson, 2013)), oxidation concentrations (e.g., photochemical age (Zhong and Jang, 2014; Kim et al., 2014; Lambe et al., 2012; Lambe et al., 2013)), and relative humidity (RH) (Titos et al., 2016; McNeill, 2015; Denjean et al., 2015; Li et al., 2017b; Sareen et al., 2017; Ye et al., 2016; Michel Flores et al., 2012) on SOA formation and the RI values of SOA derived from both biogenic and anthropogenic volatile organic compounds (VOCs). There are also many studies investigating temperature effects on SOA formation and composition (Takekawa et al., 2003; Svendby et al., 2008; Clark et al., 2016; Lamkaddam et al., 2016; Huang et al., 2018; Qing Mu and Gerhard Lammel, 2018; Zhao et al., 2019; Boyd et al., 2017; Price et al., 2016; Emanuelsson et al., 2013; Warren et al., 2009; Qi et al., 2010); however, works on the effect of temperature on the SOA RI are limited (Kim et al., 2014). Field studies have shown that temperature is an important factor affecting rate constants of the oxidation process, vapor pressure of products, and SOA formation process and yields (Atkinson and Arey, 2003; Wang et al., 2017; Roy and Choi, 2017; Ding et al., 2017; Cui et al., 2016). Therefore, investigating temperature dependence is important to our better understanding of the formation, physical, and chemical properties of SOA under tropospheric conditions.

Long-chain alkanes, an important class of intermediate-volatility organic compounds (IVOCs) (Zhao et al., 2014) and a large fraction of diesel fuel and its exhaust (Gentner et al., 2012; Gentner et al., 2017), are important potential contributor of SOA (Presto et al., 2009; Zhao et al., 2016). Several previous studies have reported the formation of SOA derived from long chain alkanes under various conditions, including SOA compositions (Fahnestock et al., 2015; Hunter et al., 2014; Lim and Ziemann, 2005), SOA yields (Loza et al., 2014; Lim and Ziemann, 2009a), and the chemical mechanisms (Yee et al., 2012, 2013; Aimanant and Ziemann, 2013; Lim and Ziemann, 2009b). Recently, Lamkaddam et al. (2016) reported the temperature dependence (10 ℃, 20 ℃, and 31.5 ℃) of SOA formation from high $NO_x$ photo-oxidaton of *n*-dodecane, and found that temperature did not significantly influence SOA yield. They attributed it to two possible reasons: the changes of reaction rate constants lead to different SOA composition; or the formed SOA are mainly non-volatile compounds that they are not sensitive to temperature. Li et al. (2017a) reported the optical properties of SOA from *n*-dodecane, *n*-pentadecane, and *n*-heptadecane

under various oxidation conditions under room temperature. However, knowledge about the effect of temperature on the chemical composition and optical properties of $n$-dodecane SOA in the absence of $NO_x$ is still lacking, which limits our understanding of the role of SOA in visibility and radiative balance under different temperatures (e.g., in winter and summer).

In the present study, we determined the temperature effects on chemical composition and optical properties of SOA generated in a smog chamber during photo-oxidation of $n$-dodecane under low-$NO_x$ condition. The results here will improve our understanding of the role of temperature in SOA chemical compositions and optical properties, and further the influence on air quality and radiative forcing.

## 2 Materials and methods

### 2.1 Smog Chamber Experiments

The experiments were performed in a dual-reactor smog chamber, the details of which were given previously (Wang et al., 2015). Briefly, the chamber consisted of two 5 m$^3$ reactors made of fluorinated ethylene propylene (FEP) Teflon-film, which were housed in a thermally isolated enclosure. The temperature in the chamber was accurately controlled by high-power air conditioner in the range of -10 - 40 °C. Multiple light sources were used in the chamber, with center wavelength of 365 nm (GE, F40BL), 340 nm (Q-lab, UVA-340), and 254 nm (PHILIPS, G36 T8). The RH and temperature in the chamber were continuously monitored and controlled during the whole experiments. The experiments were conducted under < 5% RH and under two temperatures: 25 °C (room temperature condition, R) and 5 °C (low temperature condition, L). The temperature fluctuation was ±0.5 °C for either condition. As optical properties of the SOA could be affected by many factors, in order to study the temperature affect, other factors must be kept unchanged, so the humidity of the experiments must be constant and cannot be changed. Here the experiments were conducted under dry conditions. Because when the temperature changed, the saturated vapor pressure of water changed: if the RH was consistent at different temperatures, the concentration of the water was not consistent; when the concentration of water was the same, the RH was different. So choosing other humidity (non-dry conditions) would introduce new problems, we could only choose dry conditions (RH < 5%). The $n$-dodecane (≥99%, Sigma-Aldrich) was photo-oxidized under low-NOx condition, with hydrogen peroxide (30% wt/wt, Beijing Chemical Works) as the OH precursor. $n$-Dodecane was added into the chamber first, followed by adding hydrogen peroxide. After that, wind turbine was turned on for 20 min to make sure that the materials in the chamber were well mixed.

When all the substances were added in the chamber, the chamber was set to the desired temperature. The instruments were connected at room temperature condition, when the temperature dropped to 5 °C and stabilized, the data measured by the instruments would be counted as valid data. The lights in the chamber were then turned on, and the photo-oxidation reaction started. The initial conditions for these experiments were listed in Table 1. According to our experimental design, the expected concentration of $n$-dodecane was 50 ppb, which was introducing 2 μL liquid $n$-dodecane into 5 m$^3$ smog chamber. As the injection volume of $n$-dodecane was 2 μL, volume error during injection was inevitable, which would influence the

100 concentration of *n*-dodecane in the chamber. Nevertheless, the relative small differences in *n*-dodecane concentration (43-50 ppb at low temperature and 52-58 ppb at high temperature) likely had little influence in SOA composition and optical properties.

## 2.2 Measurements

All instruments were located within one meter of the chamber, and all the connection tubes were wrapped by insulation cotton to minimize the influence of room temperature. The concentration of NOx and formed $O_3$ in the smog chamber were monitored
by the gas analyzers (Teledyne Advanced Pollution Instrumentation, Model T400 and Model T200UP, respectively). The concentration of *n*-dodecane was monitored by a proton transfer reaction quadrupole mass spectrometry (PTR-QMS 500, Ionicon), calibration of PTR-QMS's response to *n*-dodecane was achieved through permeation tubes. $NO^+$ ion source of PTR-QMS was used when detecting the *n*-dodecane (Koss et al., 2016; Shi et al., 2019; Paulsen et al., 2005) .

The particle size distribution and density were detected by a scanning mobility particle sizer (SMPS, TSI) and a centrifugal
particle mass analyzer (CPMA, Cambustion). A custom-made cavity ring-down spectrometer (CRDS) (Wang et al., 2012) was applied to monitor the optical property of the formed particles at 532 nm. A photoacoustic extinctiometer (PAX-375, Droplet Measurement Technologies) was used to measure the scattering, absorption, and extinction coefficients of formed SOA at 375 nm.

The optical properties of the formed particles were analyzed after the mass concentration of the aerosol reached the
115 maximum. During the following one to two hours, the surface mean diameter and the extinction coefficients of the particles tended to be stable and would not change much. Then, we collected the particles on the film to analyze its chemical composition in the same period. The sample collection time was chosen to make sure the signal of the collected filter is much higher than the background of the blank filter in mass spectrometry. Another principle is less sample volume used in this process. If the membrane extraction time is too long, the chamber volume will decrease too much. The formed aerosol particles were collected
on the PTFE membrane with a pore size of 200 nm (0.2 μm, 47 mm, Merckmillipore FELP). Each filter sample was collected for 30 min at 10 L/min flow rate. Then the filters were put into 5 mL methanol (99.9%, Fisher Chemical) and sonicated for 30 min. The dissolved solution was analyzed with a UV−Vis light spectrometer (Avantes 2048F), which was used to detect the absorbing property (to derive the imaginary part of RI, k) at 532 nm. The solution was also analyzed with electrospray ionization time-of-flight mass spectrometry (ESI-TOF-MS, Bruker, Impact HD) to obtain the chemical composition of the
formed SOA. Positive ion mode was used for the ESI-TOF-MS. The absolute mass error was below 3 ppm, and the typical mass resolving power is ＞30,000 at m/z 200.

## 2.3 Calculation Method of RI Values

### 2.3.1 Calculation Method of RI Values based on the CRDS

The RI values of the particles formed in the smog chamber was estimated based on both the extinction and scattering
coefficients and Mie-Lorenz theory (Bohren, 1983). The details of the calculation method of RI values are also reported in our

previous publications (Wang et al., 2012; Phillips and Smith, 2014; Li et al., 2017a; Li et al., 2017b; Li et al., 2018; Peng et al., 2018).

The extinction coefficients ($\alpha_{ext}$) of the particles with CRDS can be calculated with Equation (1):

$$\alpha_{ext} = \frac{L}{cl}\left(\frac{1}{\tau} - \frac{1}{\tau_0}\right)$$

(1)

where $L$ is the distance of the two mirrors in the cavity, $l$ is the length of the cavity that filled with aerosol particles, $c$ is the speed of the light, $\tau_0$ is the ring down time of the CRDS when it is filled with zero air, and $\tau$ is the ting down time of the CRDS when it is filled with aerosol particles

For the homogeneous spherical particles, the $\alpha_{ext}$ can be calculated with Equation (2):

$$\alpha_{ext} = N\sigma_{ext} = \frac{1}{4}N\pi D^2 Q_{ext}$$

(2)

where $N$ is the concentration of the spherical particles, $\sigma_{ext}$ is the extinction cross section, $D$ is the particle diameter, and $Q_{ext}$ is the extinction efficiency.

$Q_{ext}$ is the ratio of Beer's law extinction cross section to the geometric area of the spherical particles, it is dimensionless,

and can be expressed with Equation (3):

$$Q_{ext} = \frac{4\alpha_{ext}}{N\pi D^2}$$ (3)

For the polydisperse SOA particles formed in the smog chamber, they follow log-normal distribution and the geometric standard deviation is always smaller than 1.5, the $Q_{ext}$ with a surface mean diameter $D_s$ can be expressed with Equation (4) (with the assumption that the aerosol particles during each size bin are homogeneous spherical particles) (Nakayama et al.,

2010):

$$Q_{ext}(D_s) = \frac{\alpha_{ext}}{\int N(D_p)\frac{1}{4}\pi D_p^2 dD_p}$$ (4)

where $D_p$ is the geometrical diameter of the particle, $dD_p$ is the size bin of the particles, and $N(D_p)$ is the number concentration of the particles in $dD_p$ with $D_p$ per unit volume.

For the particles with $D_p$ in each size bin $dD_p$, the surface area $S(D_p)$ can be expressed by Equation (5):

$$S(D_P) = N(D_p)\pi D_p^2$$ (5)

So the Equation (4) can also be expressed with Eauqtion (6):

$$Q_{ext}(D_s) = \frac{\alpha_{ext}}{\int\frac{1}{4}S(D_p)dD_p} = \frac{4\alpha_{ext}}{S_{tot}}$$

(6)

where $S_{tot}$ is the total surface area of the particles, and the values can be obtained with SMPS.

While the extinction efficiency can also be calculated with Mie-Lorenz theory, Equation (7):

$$Q_{ext}(D_s) = \int f(D_p)Q_{ext}(D_p)dD_p$$

(7)

where $f(D_p)$ is the normalized surface area weighted size distribution function.

Then the measured extinction efficiency ($Q_{ext,mea}$) is compared to calculated extinction efficiency ($Q_{ext,cal}$). And the best-fit RI value is determined by minimizing the following reduced merit function ($\chi_r$), Equation (8):

$$\chi_r = \frac{1}{N}\sum_{i=1}^{N}(Q_{ext,mea} - Q_{ext,cal}(n,k))_i^2$$

(8)

where $N$ is the number of diameters to be calculated.

The uncertainties of the particle concentration and surface mean diameter measured by SMPS are $\pm$ 10% and $\pm$ 1% respectively. The uncertainty of the retrieval method is $\pm$ 0.002, and the uncertainty of the measured extinction coefficient with CRDS is $\pm$ 3%, resulting in the final uncertainty of the retrieved RI value to be about 0.02-0.03. And the corresponding equation for the RI value uncertainty can be referred to the Supporting Information.

### 2.3.2 Calculation Method of RI Values based on QSPR

Using the molecular formula obtained from ESI-TOF-MS, we calculated the RI values of the products in SOA with the quantitative structure–property relationship (QSPR) method (Redmond and Thompson, 2011). The QSPR can be expressed with Equation (9):

$$RI_{predicted} = 0.031717(\mu) + 0.0006087(\alpha) - 3.0227\left(\frac{\rho_m}{M}\right) + 1.38709 \qquad (9)$$

where $\mu$ is the unsaturation of the molecular formula, $\alpha$ is the polarizability of the molecular formula, $\rho_m$ is the mass density (g/cm$^3$), and $M$ is the molar mass (g/mol).

The mass density of the compound is estimated by AIM model, the detailes of which can be referred to Girolami (1994).

$\mu$ is calculated through the conventional approach, which is used in many organic chemistry texts, Equation (10)

$$\mu = (\#C + 1) - 0.5(\#H - \#N) \qquad (10)$$

where #C, #H, and #N are the number of the C, H, and N respectively.

$\alpha$ is calculated based on the molecular formula of the compound, it can be expressed by Equation (11):

$$\alpha = 1.51(\#C) + 0.17(\#H) + 0.57(\#O) + 1.05(\#N) + 2.99(\#S) + 2.48(\#P) + 0.22(\#F) + 2.16(\#Cl) +$$
$$3.29(\#Br) + 5.45(\#I) + 0.32 \qquad (11)$$

where # is the number of the atoms of each element in the molecular formula.

The calculated RI values of products were used to link chemical composition and optical properties, and to explain the obseved RI differences at different temperatures in Sect. 3.4.

### 2.3.3 Calculation Method of Average Elemental Composition and Ratios

Due to the complexity of chemical composition in secondary organic aerosol particles, it is common to express the bulk composition as averaged elemental ratios. The average elemental composition and ratios (C, H, O, H/C, O/C) can be estimated from the identified molecular formulas (Ervens et al., 2011):

$$< Y > = \frac{\sum_i x_i Y_i}{\sum_i x_i} \tag{12}$$

$$< Y/Z > = \frac{\sum_i x_i Y_i}{\sum_i x_i Z_i} \tag{13}$$

where Y=C, H, O, Y/Z=H/C, O/C. $x_i$ is the peak abundance of elemental composition.

### 2.4 Impact of Temperature on Direct Radiative Forcing

The simple forcing efficiency (SFE) is used to dertermine the relative importance of optical properties of the aerosol to direct radiative forcing at the Earth's surface (Bond and Bergstrom, 2006):

$$SFE = \frac{S_0}{4} \tau_{atm}^2 (1 - F_c) [2(1 - a_s)^2 \frac{Q_{bs}C}{M} - 4a_s \frac{Q_a C}{M}] \tag{14}$$

where $S_0$ is the solar radiation, $\tau_{atm}$ is the transmission of the atmosphere, $F_c$ is the cloud fraction, $a_s$ is the surface albedo, $Q_{bs}$ and $Q_a$ are the backscattering and absorption efficiency of the aerosol particles, $M$ is the aerosol mass, and $C$ is the cross section of the aerosol.

The SOA derived from $n$-dodecane have negligible absorption at the wavelengths of 532 nm and 375 nm under the two temperature conditions, so the value of $Q_a$ is zero, and the impact of temperature on the direct radiative forcing (DRF) can be expressed with Eq. (15), and this will be discussed Section. 3.5.

$$DRF \ ratio = \frac{SFE_{low \ temperature}}{SFE_{normal \ temperature}} = \frac{Q_{bs,low \ tem.}}{Q_{bs,normal \ tem.}} \tag{15}$$

## 3. Results and Discussion

### 3.1 Photo-oxidation Experiments

The profiles of the $n$-dodecane photo-oxidation experiments at different temperatures were shown in Figure S1. As similar amount of $n$-dodecane and oxidant were added to the smog chamber and the reaction rate was lower under low temperature, the reaction time of low temperature condition was significantly longer than the room temperature condition: 4 to 5 h for room temperature, and 8 to 9 h for low temperature. Under room temperature condition, the total surface concentration and mass reached a maximum after 3 h; while for low temperature condition, this time was 6 h.

## 3.2 Effect of Temperature on RI

Optical properties of the formed particles were analyzed after the mass concentration of the aerosol reached the maximum, and the last 1 h data were used. During this period, the surface mean diameter and the extinction coefficients ($\alpha_{ext}$) of the particles tended to be stable and would not change much. The extinction coefficients and extinction efficiency of the particles as a function of the surface mean diameter were shown in Figure S4. The SOA derived from *n*-dodecane had no significant absorption both at 532 nm and 375 nm, similar to our previous study (Li et al., 2017a), and the imaginary part of RI will not be discussed here. The real part of RI obtained in this study was shown in Figure 1. As shown in Figure 1, the RI values at 532 nm under room temperature were similar to our previous study (Li et al., 2017a), while the two temperature conditions had significantly different ranges for RI: 1.472-1.486 (25 ˚C) and 1.502-1.526 (5 ˚C) at 532 nm; 1.51-1.53 (25 ˚C) and 1.532-1.56 (5 ˚C) at 375 nm. And the details of the variation tendency of the RI values as a function of surface mean diameter for SOA produced under room and low temperature conditions were shown in Figure 2. The various RI values at different temperature indicated that lower reaction temperature (from 25 ˚C to 5 ˚C) had an enhance effect (~0.03 at 532 nm, ~0.02 at 375 nm) on the RI of *n*-dodecane SOA. The mass spectrometry analysis below (Section 3.3) was applied to obtain chemical composition information and explain the phenomenon above.

## 3.3 Temperature Effect on Chemical Composition and Reaction Mechanism

The higher RI values under low temperature indicated that the temperature might change the chemical composition of SOA by changing the reaction types or shifting the balance of different pathways. The mass spectra (MS) of *n*-dodecane SOA obtained by ESI-TOF-MS in positive ion mode were shown in Figure 3, which provided the molecular insight into the chemical changes under different temperature conditions. We identified about 260 individual masses for SOA under two temperature conditions, and the details were shown in Figure S2 and Table S1. The spectrum under low temperature was significantly different with the spectrum under room temperature, with large amount of ions corresponding to monomer, dimer, trimer, and tetramer. This suggested that oligomerization might play a dominant role under low temperature condition.

The molecular composition of *n*-dodecane SOA was significantly modified by temperature conditions, with the averaging SOA formula changing from $C_{14.98}H_{26.47}O_{5.53}$ (R) to $C_{21.25}H_{40.44}O_{7.43}$ (L). The average carbon number increased from 14.98 at room temperature to 21.25 at low temperature, indicating that the SOA molecules were larger at low temperature. The average O/C and H/C ratios at room temperature condition were 0.37 and 1.72, respectively, while at low temperature condition the ratios were 0.35 and 1.90, respectively. The products formed under low temperature tended to have higher H/C ratio and lower O/C ratio compared with the products under room temperature. The details of the O/C and H/C ratios of the products formed at different temperature conditions can be referred to Figure 5a. The phenomenon above might be due to the presence of oligomers.

For the oligomers in the mass spectrum, one possible explanation was that low temperature condition promoted gas-particle partitioning and changed the particle phase reaction. The gas-phase OH oxidation could reduce H/C and increase O/C

ratios (Li et al., 2019; Lambe et al., 2015; Li et al., 2018), while particle-phase oligomerization almost won't change O/C and H/C ratios (Charron et al., 2019). Under room temperature condition, more gas-phase oxidation steps were needed to produce the less volatile products to condense into particle-phase (because of the high temperature, i.e., high saturation vapor pressure). Hence, products formed under room temperature had lower H/C and higher O/C ratios. In contrast, the more volatile (i.e., less oxidized) products were readily to condense into particle phase under low temperature, and then undergo particle phase reactions, e.g., oligomerization, leading to the formation of products with higher H/C and lower O/C ratios. This phenomenon was consistent with Kim et al. (2014), they studied the dependence of real part of RI on O/C and H/C ratios of SOA derived from limonene and α-pinene, and found that the higher percentage of less oxygenated semivolatile substances were responsible for the higher RI values.

For particle-phase reactions, there were mainly two reaction pathways (Fahnestock et al., 2015; Yee et al., 2012, 2013): intramolecular cyclization of multifunctional hydroperoxides (form furan derivatives); intermolecular oligomerization of multifunctional hydroperoxides with aldehydes (form peroxyhemiacetal, PHA). The two pathways were competitive during the particle phase reaction process. According to the mass spectra analysis, we speculated that the low temperature promoted the progress of the oligomerization reaction, and made it the primary pathway in the particle phase, as shown in Figure 4. As discussed above, the products with higher volatility and lower oxidation state would condense on the particle phase under low temperature condition and then participate in the particle phase reaction, which would further promote the oligomerization reaction.

Another possible explanation was that the pathway of the gas phase reaction changed under low temperature condition comparing with room temperature condition. Combining existing mass spectrometry information, we speculated that under low temperature condition the oligomers might also be formed by gas phase radical oligomerization and then rapidly deposited into the particle phase. However, these are speculations based on the existing analysis results. The specific reaction mechanism under low temperature condition needs further investigation.

Clark et al. (2016) found that the SOA formed from isoprene had similar mass spectra at the monomer range at different temperatures. However, the precursors are very different (isoprene vs $n$-dodecane), which have very different oxidation pathways and partitioning processes. Hence, it is possible that we observe different results in mass spectra at monomer range. As we have mentioned above, the different oxidation degree and partitioning process have made the difference in monomer range. At room temperature, more OH oxidation steps in the gas phase can lead to the formation of some fragmentation products that may not be observed in the low temperature. For example, we have observed the high intensity of M/Z 195 $C_{10}H_{20}O_2$ (relative intensity 0.39), M/Z 211 $C_{10}H_{20}O_3$ (relative intensity 0.13) at room temperature but very low intensity (0.012, 0.041, respectively) at low temperature, and M/Z 195 and 211 are likely products from gas-phase OH oxidation. This process may be different from isoprene (Clark et al., 2016), because for $n$-dodecane the carbon number is high and volatility is low and fewer oxidation steps are needed before partitioning, while for isoprene there is only 5 carbon atoms so more oxidation steps are needed before partitioning no matter at room or low temperature. Overall, the differences in monomer range still make sense.

### 3.4 Relationship between RIs and Chemical Composition of SOA

Refractive index of aerosol particles is fundamentally the results of a combination of particle chemical compositions and internal mixing. SOA particles formed in the smog chamber are treated as homogenous mixtures, and the RI values of which can be expressed as (Redmond and Thompson, 2011):

$$RI = \sum x_i RI_i \tag{16}$$

where $x_i$ is the fraction of the $i_{th}$ component, and $RI_i$ is the refractive index of component i.

The RI values were calculated for the identified products (with high intensity) under room and low temperature conditions, and the details were shown in Figure 5b,c and Table S1. It could be clearly seen that the RI values of the products at room temperature condition were mainly in the range of 1.4 to 1.5, and the degree of oligomerization was mainly in the range of monomer and dimerization. However, for the products under low temperature condition, the RI value was in the range of 1.4 to 1.55, and the degree of oligomers could reach tetramerization. As the degree of oligomerization increased, RI was gradually increasing as well. This generally explained the higher RI of SOA under lower temperature condition.

To further validate our speculation and identify the relationship between the measured RI values and the chemical composition of *n*-dodecane SOA, we chosed a surrogate system containing 11 PHA oligomerization and 2 cyclization reactions (with different degree of unsaturation and functional groups) to calculate the expected RI values using the quantitative structure−property relationship (QSPR) method based on the molecular formula and structure (Redmond and Thompson, 2011). The details of the 13 reactions and related molecular information were shown in Figure S3 and Table 2, while the relationship between the predicted RI values with the degree of unsaturation, and the degree of oligomerization were shown in Figure 6. Strong correlations were observed between predicted RI values and unsaturation, and the degree of oligomerization. With the same unsaturation, the RI values would increase with the increasing degree of oligomerization; under the same degree oligomerization, the RI values would increase with the increasing degree of unsaturation. In addition, the RI values of the surrogate system were similar with the identified substances, which further confirmed the effect of oligomers on RI values.

In addition to the oligomer reasons above, the difference in wall losses and gas-particle partitioning of gas-phase products might partially contribute to the RI enhancement under low temperature condition. The particle wall loss rates under both room and low temperature conditions had been measured. The wall loss rate under low temperature condition (0.0025 ~ 0.0028 min$^{-1}$) was larger than that under room temperature condition (0.0018 ~ 0.0020 min$^{-1}$). After wall loss correction, the SOA mass under low temperature condition was higher than that under room temperature condition. However, this difference in particle wall loss rate can only slightly change the SOA mass concentration, but not the particle chemical composition. Therefore, it is unlikely to change the optical properties. Nevertheless, the difference in vapor wall loss rates may change the particle composition and optical properties. The low temperature can enhance the loss rates of higher-volatility compounds (while for lower-volatility compounds, their dominant fates are condensation so temperature may affect little on their losses), which may lead to their lower proportions in SOA particles. As their RIs are generally lower than lower-volatility compounds (Li et al., 2018), this proportion change can probably enhance SOA RI at low temperature. Therefore, the difference in wall losses and

gas-particle partitioning of gas-phase products might partially contribute to the RI enhancement under low temperature condition.

## 3.5 Atmospheric and Climate Implications

Figure 7 showed the ratios of light extinction efficiency ($Q_{ext}$) and direct radiative forcing (DRF) of *n*-dodecane SOA under different temperature conditions, from which we could know the impacts of temperature on the role of *n*-dodecane SOA in visibility and radiative balance. As shown in Figure 7a and c, the extinction efficiency ($Q_{ext}$) of SOA generated under low temperature is larger than SOA generated under room temperature in the size range of 50 to 200 nm, the range of which are the most atmospherically relevant (Zhang et al., 2015; Guo et al., 2014). The enhancement is about 7%-20% at 532 nm, and about 1%-21% at 375 nm. This suggests that the extinction efficiency ($Q_{ext}$) of SOA formed from *n*-dodecane (perhaps other long-chain alkanes in general) was higher in winter than in summer, which would result in lower visibility.

The enhancement in light extinction of SOA and oligomer compositions formed under low temperature condition might provide some possible inspiration for the regional visibility issues, especially the suburban areas. It had been reported that UV-scattering particles in the boundary layer could accelerate photochemical reactions and haze production (Sun et al., 2014). The observations above showed that the scattering property of formed SOA increased under low temperature condition, which might provide one possible reason for the rapid occurrence of haze in suburban areas in winter. According to field observations, haze occurred frequently in winter, especially in China (Cheng et al., 2016; Huang et al., 2014; Guo et al., 2014; Parrish et al., 2007).When haze occurred, it was often accompanied by high $NO_X$, especially in urban areas. The temperature effects under high-$NO_X$ conditions are also important and need to be investigated in future studies.

As shown in Figure 7b and d, the DRF under low temperature condition was generally larger than room temperature condition: the enhancement was about 6%-19% at 532 nm (50-200 nm), and about 7%-22% at 375 nm (50-180 nm); while for the size range of 180-200 nm, low temperature decreased the DRF ratio about 3% at 375 nm. This phenomenon implied that the SOA generated in winter (low temperature) might have larger DRF on the Earth's surface than the SOA generated in summer (room temperature). It may also imply that the temperature condition should be considered when evaluating the DRF of the aerosol particles generated in the atmosphere.

## 4 Conclusions

To the best of our knowledge, this was the first report about optical properties of long-chain alkane SOA at low temperature condition. The modification in temperature significantly changed the chemical composition of the particulate phase. From the oligomer component at low temperature condition, it was presumed that the oligomerization was dominant at low temperature. The presence of oligomers in the SOA particles resulted in an increase of RI values. This study would help to improve our understanding of the lower visibility and the formation of haze in winter. Our results also showed the need for further investigation on the atmospheric parameters influencing SOA formation and optical properties.

**Author contributions.** WW conceived and led the studies. JL,WZ, and CP performed chamber simulation and data analysis. KL, LZ, BS, YC, ML, HL and MG discussed the results and commented on the manuscript. JL prepared the manuscript with
contributions from all co-authors.

**Data availability.** The data used in this study are available upon request from the corresponding author.

**Competing interests.** The authors declare that they have no conflict of interest.

**Acknowledgements**

This project was supported by the National Key Research and Development Program of China (2017YFC0209506) and National Natural Science Foundation of China (41822703, 91744204, 91844301).

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

**Table 1: The initial conditions of the smog-chamber experiments.**

| Experiments No.[a] | HC (ppb) | H$_2$O$_2$ (ppm) | NO$_X$ (ppb) | RH (%) | Temperature (℃) | Mass[b] (µg/m³) | Mass[c] (µg/m³) |
|---|---|---|---|---|---|---|---|
| Dod-R-1 | 58 | 1.03 | <1 | <5 | 25 | 155 | 225 |
| Dod-R-2 | 52 | 1.03 | <1 | <5 | 25 | 135 | 197 |
| Dod-L-1 | 43 | 1.07 | <1 | <5 | 5 | 128 | 342 |
| Dod-L-2 | 50 | 1.09 | <1 | <5 | 5 | 133 | 387 |

[a] Experimental conditions: *n*-Dodecane room temperature (Dod-R), *n*-Dodecane low temperature (Dod-L)

[b] The mass concentration is obtained without wall loss correction.

[c] The mass concentration is calculated with wall loss correction.

**Table 2: The calculated RI values for the surrogate system: oligomers (11 types of PHA) and furan derivatives.**

| Molecular Formula | Molecular Weight | Polarizability | Unsaturation | Predicted RI |
|---|---|---|---|---|
| C$_{12}$H$_{26}$O$_3$ | 218.32 | 24.57 | 0 | 1.39017 |
| C$_{12}$H$_{24}$O$_3$ | 216.31 | 24.23 | 1 | 1.42236 |
| C$_{12}$H$_{22}$O$_4$ | 230.29 | 24.46 | 2 | 1.45422 |
| C$_{12}$H$_{24}$O$_4$ | 232.31 | 24.8 | 1 | 1.42203 |
| C$_{12}$H$_{22}$O$_5$ | 246.29 | 25.03 | 2 | 1.45389 |
| C$_{12}$H$_{24}$O$_5$ | 248.3 | 25.37 | 1 | 1.42174 |

| | | | | |
|---|---|---|---|---|
| $C_{12}H_{20}O_5$ | 244.27 | 24.69 | 3 | 1.48607 |
| $C_{12}H_{18}O_6$ | 258.26 | 24.92 | 4 | 1.51681 |
| $C_{12}H_{20}O_6$ | 260.27 | 25.26 | 3 | 1.48574 |
| $C_{12}H_{24}O_6$ | 264.3 | 25.94 | 1 | 1.4215 |
| $C_{12}H_{22}O_6$ | 262.29 | 25.6 | 2 | 1.4536 |
| $C_{22}H_{44}O_6$ | 404.56 | 44.44 | 1 | 1.43921 |
| $C_{22}H_{46}O_6$ | 406.58 | 44.78 | 0 | 1.40724 |
| $C_{22}H_{42}O_7$ | 418.54 | 44.67 | 2 | 1.47106 |
| $C_{22}H_{44}O_7$ | 420.56 | 45.01 | 1 | 1.43909 |
| $C_{22}H_{42}O_8$ | 434.54 | 45.24 | 2 | 1.47095 |
| $C_{22}H_{44}O_8$ | 436.56 | 45.58 | 1 | 1.43899 |
| $C_{22}H_{40}O_8$ | 432.53 | 44.9 | 3 | 1.50292 |
| $C_{22}H_{38}O_9$ | 446.51 | 45.13 | 4 | 1.53478 |
| $C_{22}H_{40}O_9$ | 448.53 | 45.47 | 3 | 1.50281 |
| $C_{22}H_{44}O_9$ | 452.56 | 46.15 | 1 | 1.43948 |
| $C_{22}H_{42}O_9$ | 450.54 | 45.81 | 2 | 1.47085 |
| $C_{32}H_{64}O_9$ | 592.82 | 64.65 | 1 | 1.45319 |
| $C_{32}H_{66}O_9$ | 594.83 | 64.99 | 0 | 1.42134 |
| $C_{32}H_{62}O_{10}$ | 606.8 | 64.88 | 2 | 1.48505 |
| $C_{32}H_{64}O_{10}$ | 608.82 | 65.22 | 1 | 1.4532 |
| $C_{32}H_{62}O_{11}$ | 622.8 | 65.45 | 2 | 1.48505 |
| $C_{32}H_{64}O_{11}$ | 624.81 | 65.79 | 1 | 1.45361 |
| $C_{32}H_{60}O_{11}$ | 620.78 | 65.11 | 3 | 1.5169 |
| $C_{32}H_{58}O_{12}$ | 634.77 | 65.34 | 4 | 1.54876 |
| $C_{32}H_{60}O_{12}$ | 637.78 | 65.68 | 3 | 1.51691 |
| $C_{32}H_{64}O_{12}$ | 640.81 | 66.36 | 1 | 1.45403 |
| $C_{32}H_{62}O_{12}$ | 638.8 | 66.02 | 2 | 1.48547 |
| $C_{42}H_{84}O_{12}$ | 781.07 | 84.86 | 1 | 1.46637 |
| $C_{42}H_{86}O_{12}$ | 783.09 | 85.2 | 0 | 1.4349 |
| $C_{42}H_{82}O_{13}$ | 795.05 | 85.09 | 2 | 1.49823 |
| $C_{42}H_{84}O_{13}$ | 797.07 | 85.43 | 1 | 1.46676 |
| $C_{42}H_{82}O_{14}$ | 811.05 | 85.66 | 2 | 1.49861 |
| $C_{42}H_{84}O_{14}$ | 813.07 | 86 | 1 | 1.46715 |
| $C_{42}H_{80}O_{14}$ | 809.04 | 85.32 | 3 | 1.53008 |
| $C_{42}H_{78}O_{15}$ | 823.02 | 85.55 | 4 | 1.56194 |
| $C_{42}H_{80}O_{15}$ | 825.04 | 85.89 | 3 | 1.53047 |

| | | | | |
|---|---|---|---|---|
| $C_{42}H_{84}O_{15}$ | 829.07 | 86.57 | 1 | 1.46753 |
| $C_{42}H_{82}O_{15}$ | 827.05 | 86.23 | 2 | 1.499 |
| $C_{12}H_{23}O_2$ | 199.3 | 23.49 | 1.5 | 1.43592 |
| $C_{12}H_{21}O_3$ | 213.28 | 23.72 | 2.5 | 1.46778 |
| $C_{12}H_{23}O_3$ | 215.3 | 24.06 | 1.5 | 1.43562 |
| $C_{12}H_{22}O_3$ | 214.29 | 23.89 | 2 | 1.45228 |
| $C_{12}H_{21}O_4$ | 229.28 | 24.29 | 2.5 | 1.46748 |
| $C_{12}H_{20}O_4$ | 228.27 | 24.12 | 3 | 1.48413 |
| $C_{12}H_{22}O_4$ | 230.29 | 24.46 | 2 | 1.45198 |
| $C_{12}H_{19}O_4$ | 227.27 | 23.95 | 3.5 | 1.49964 |
| $C_{12}H_{17}O_5$ | 241.25 | 24.18 | 4.5 | 1.5315 |
| $C_{12}H_{19}O_5$ | 243.27 | 24.52 | 3.5 | 1.49934 |
| $C_{12}H_{18}O_5$ | 242.26 | 24.35 | 4 | 1.51599 |
| $C_{12}H_{22}O_5$ | 246.29 | 25.03 | 2 | 1.45173 |
| $C_{12}H_{21}O_5$ | 245.28 | 24.86 | 2.5 | 1.46723 |
| $C_{12}H_{20}O_5$ | 244.27 | 24.69 | 3 | 1.48384 |

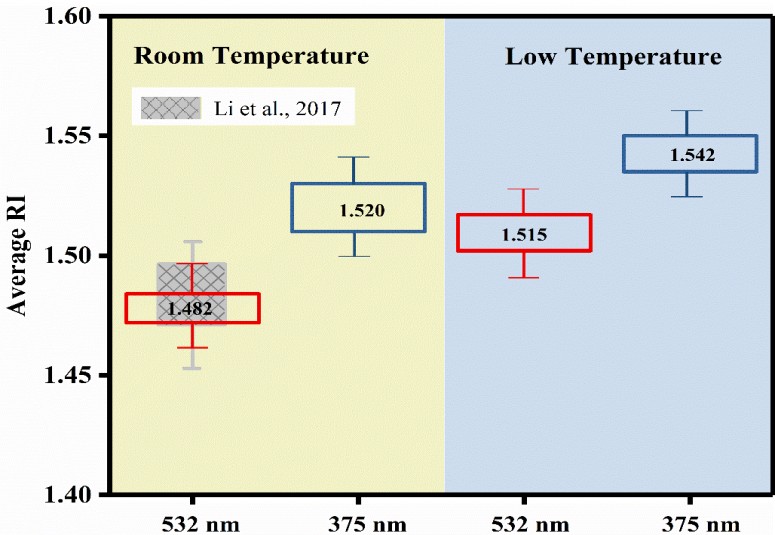

**Figure 1: Summary of the averaged RI values of dodecane SOA under room temperature 25 ˚C (the light orange area) and low temperature 5 ˚C (the light blue area) in the wavelength of 532 nm and 375 nm. The red box is the averaged RI value for *n*-dodecane in 532 nm, the shaded boxes are the RI values from our previous study in 532 nm (Li et al., 2017a), the blue box is the averaged RI value for *n*-dodecane in 375 nm.**

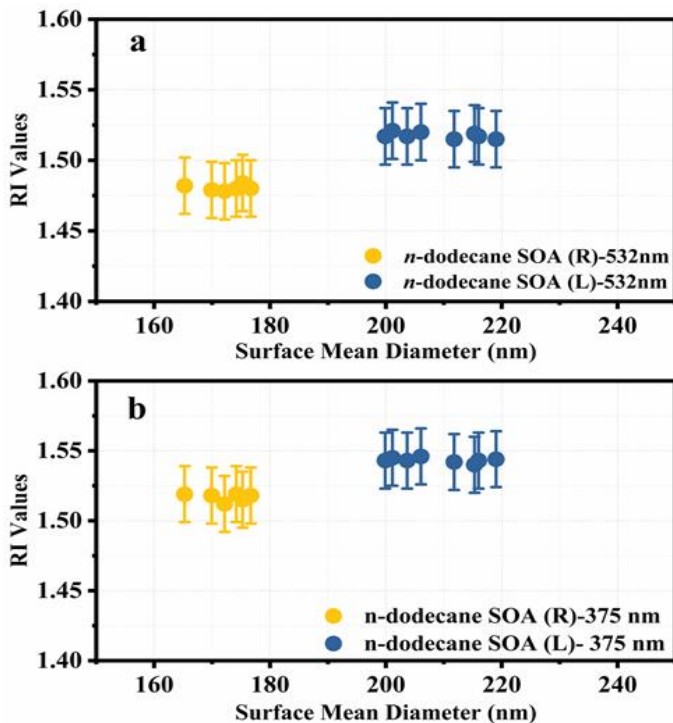

**Figure 2: Variation tendency of the RI values as a function of surface mean diameter for SOA produced under room and low temperature conditions at (a) 532 nm and (b) 375 nm.**

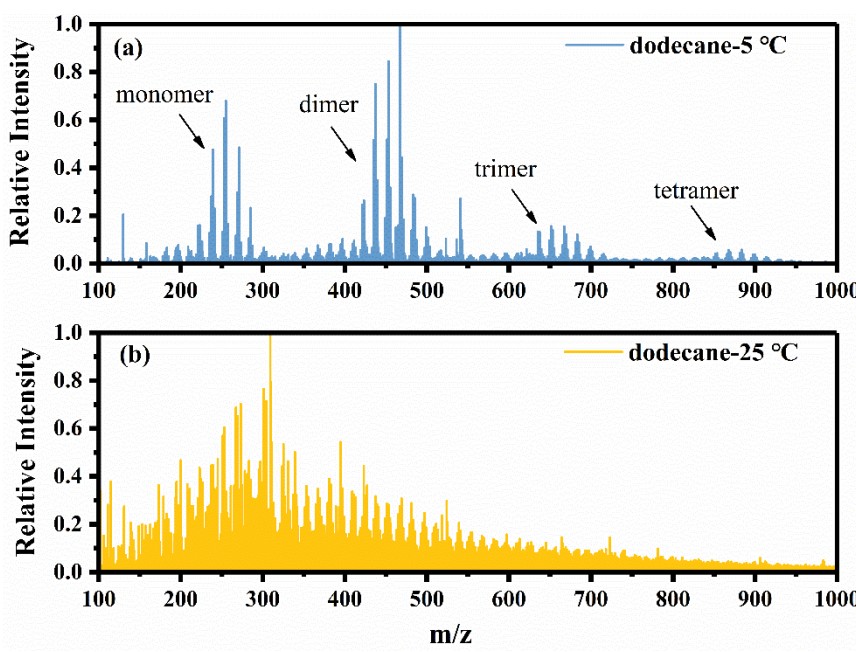

**Figure 3: Mass spectra of *n*-dodecane SOA obtained by ESI-TOF-MS in positive ion mode. (a) low temperature condition; (b) room temperature condition.**

**Figure 4: The proposed reaction mechanism of long-chain alkanes under low-NOx condition (Fahnestock et al., 2015;Yee et al., 2013, 2012)**

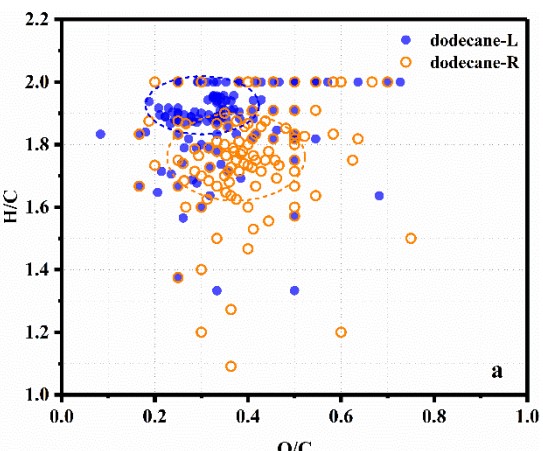 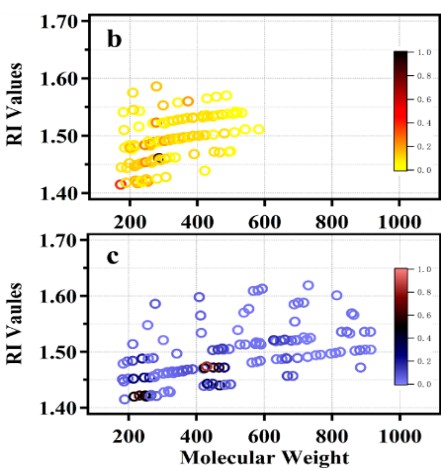

**Figure 5: (a) Van Krevelen plots showing O/C and H/C ratios for identified products by EST-TOF-MS: the orange circle is for the SOA generated under room temperature condition; the blue circle is for the SOA generated under low temperature condition; (b) calculated RI values of the selected identified molecular species from MS spectra under room temperature, the color map refers to the relative intensity of the molecular formula; (c) calculated RI values of the selected identified molecular species from MS spectra under low temperature, the color map refers to the relative intensity of the molecular formula.**

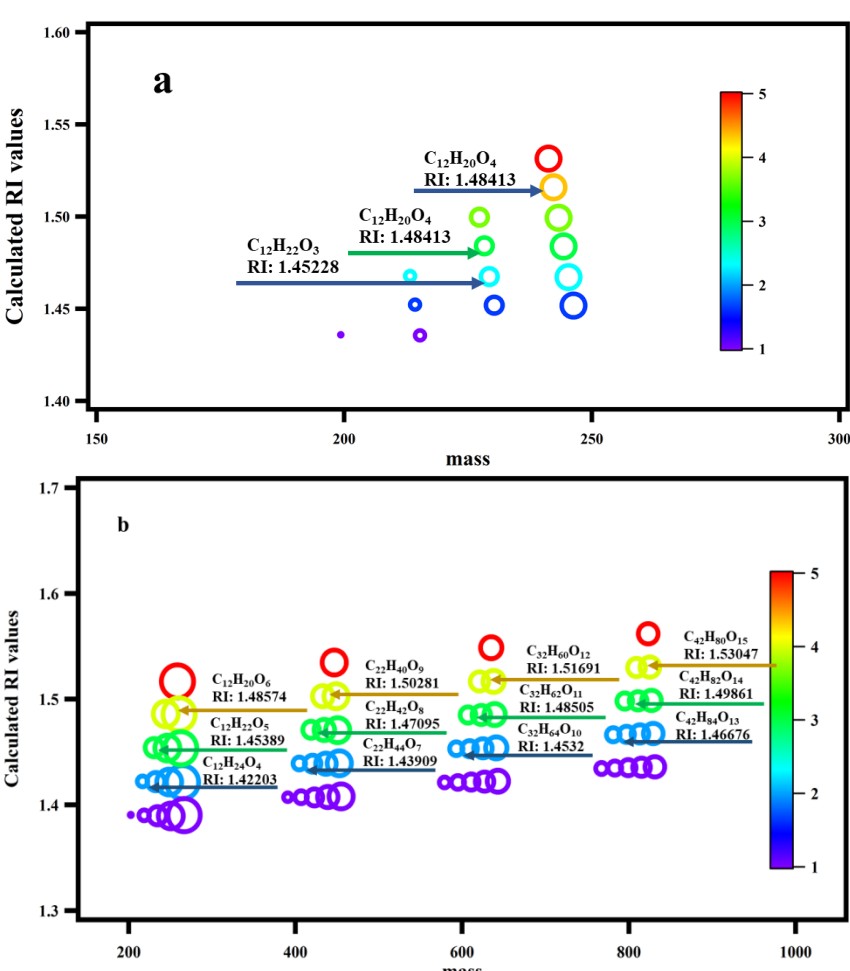

**Figure 6: Calculated RIs values of the surrogate system: (a) furan derivatives via intramolecular reaction (b) the oligomers via the formation of peroxy-hemiacetal. The color map refers to the unsaturation of the molecular formula. The size of the circle refers to the O/C ratio.**

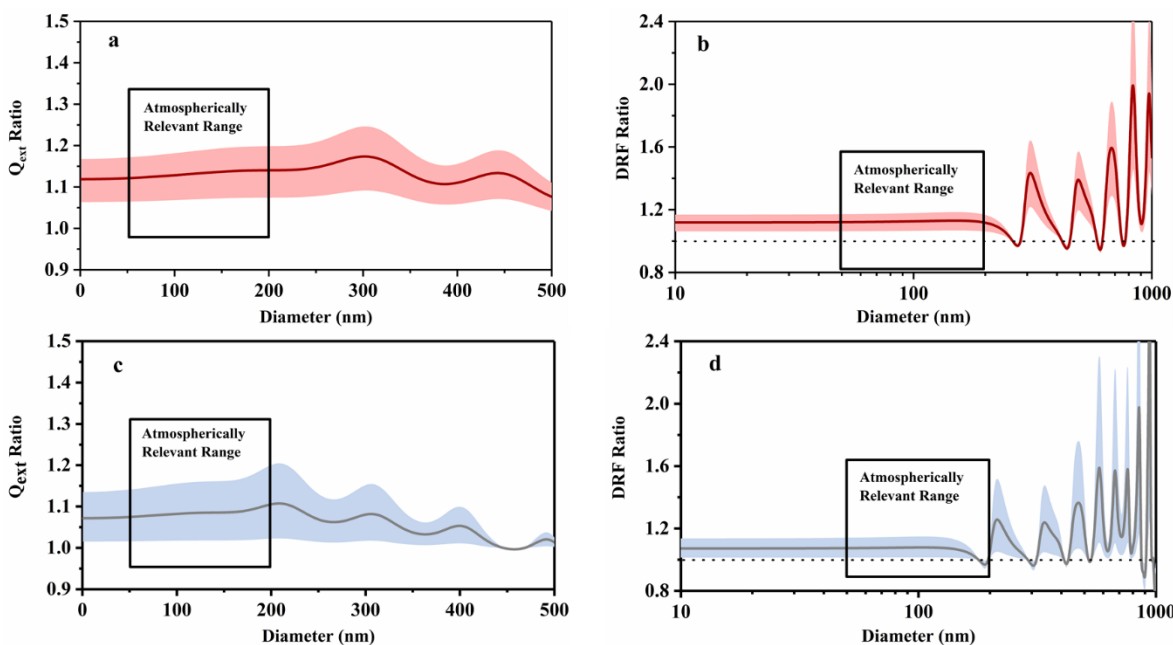

**Figure 7: The ratio of light extinction efficiency ($Q_{ext}$ ratio) and direct radiative forcing (DRF ratio) of *n*-dodecane SOA under different temperature conditions. (a) $Q_{ext}$ ratio at 532 nm; (b) DRF ratio at 532 nm; (c) $Q_{ext}$ ratio at 375 nm; (d) DRF ratio at 375 nm. The solid line is the average value of the ratio, and the shaded area is the uncertainty.**