# Peer review of "Temperature Effects on Optical Properties and Chemical Composition of Secondary Organic Aerosol Derived from *n*-Dodecane"

_Atmospheric Chemistry and Physics, 2019_

## Referee Comment (RC1) · Anonymous Referee #1 · 8 Mar 2020

Li et al. presented a study that examined the effects of temperature on the optical properties and chemical composition of secondary organic aerosols formed from the OH photooxidation of n-dodecane. The authors found that oligomers were formed at low temperatures, and these oligomers resulted in higher RI values being measured. This paper is potentially useful to the SOA community. However, there are some important issues that the authors need to address before the manuscript can be considered for publication.

Major comments: 1. Why were experiments conducted under dry conditions? 2. Table 1 showed used of 43 ppb at low temperature vs. 58 ppb at high temperature. Why

wasn't the same amount of n-dodecane used? 3. Do the authors know how the use of different temperatures will affect the loss rates of particles to the chamber walls? Is there a possibility that the observations of the SOA mass, composition and optical properties made by the authors can be explained partly by differences of particle wall loss rates at different temperatures? 4. In page 5 line 144, the authors state that "During this period, the optical properties of the particles tend to be stable and will not change much." This sentence is ambiguous and needs to be clarified. What optical property is the authors referring to? RI value? Or are they referring to the absorption spectra? 5. Explain the rationale behind tracking the absorption at 532 and 375 nm. 6. The authors did not use seed aerosols in this study to promote gas-to-aerosol partitioning. Hence, I expect substantial vapor wall loss in these experiments, and the extent of vapor wall loss is likely to be different at 5 C vs. 25 C. Is it possible that the authors are not detecting some products (due to their loss to the chamber walls) that can contribute to SOA optical properties? 7. In section 3.5, the authors tried to relate their results to observations made during winter haze episodes in China. I advise the authors to be more circumspect in the extrapolation of their results to ambient observations since NOx concentrations are likely substantial during winter haze episodes in China. The authors performed a study under low-NOx conditions. Under high NOx conditions, I expect the reaction mechanism of n-dodecane OH photooxidation to be different. For example, more fragmentation will likely happen, which will result in the formation of more volatile products. If this is the case for both 5 C and 25 C conditions, there may not be significant differences between SOA composition and their RI values under high NOx conditions, which would imply that temperature does not play a big role in the DRF of SOA formed under areas with significant NOx concentrations, like China.

Minor comments: 8. Inconsistent tenses. The authors switch between using past and present tenses in some parts of the manuscript. Please correct this.
* * *
[Figure]

2020.

---

## Referee Comment (RC2) · Anonymous Referee #2 · 14 Mar 2020

General comments: This work by Li et al. describes measurements of optical properties in relation with chemical composition of n-dodecane SOA under low-NOx and two different temperatures (5ïĆřC and 25ïĆřC). The authors found that under low temperature, the real part of the refractive index (RI) at 375 nm and 532 nm is enhanced, corresponding to substantial oligomer formation. The authors hence conclude that the enhanced oligomer formation under low temperature lead to the higher RI. The results could be relevant to low visibility in urban areas during wintertime. Overall, the manuscript is well written and demonstrates new findings regarding SOA optical properties vs. chemical compositions. But a few major concerns need to be addressed before this manuscript can be considered publishing.

[Figure]

Specific comments: 1. Line 49 – 52. Although it is true that few studies have examined temperature effects on RI values of SOA, temperature effect studies on SOA formation and composition are not limited. There are many other studies on SOA formation and temperature effects not cited. For example, Warren et al., 2009, 43, 3548, Atmos. Environ.; Emanuelsson et al., 2013, 117, 10346, J. Phys. Chem. A.; Price et al., 2016, 50, 1216, Aerosol. Sci. Technol.; Boyd et al., 2016, 51, 7831, Environ. Sci. Technol.; Zhao et al., 2019, 3, 2549, ACS Earth Space Chem., etc.

2. Line 78. Although details of the chamber can be referred to previous work, important and fundamental characteristics of the chamber still needs to be provided. For example, the material and volume of the chamber.

3. Some important details of the chamber experiments are missing. Why were instruments not connected before temperature was stabilized? SOA mass concentration in each experiment should be reported (in Table 1 or in the main text). How long were filter samples collected and at what flow rate? After the filters were dissolved in filter, were the solutions sonicated? If not, how good were the extraction recoveries?

4. How did the extinction coefficient and extinction efficiency evolve over the course of an experiment?

5. Line 122. It is unclear how was the RI value uncertainty estimated from the various uncertainties. An equation needs to be provided.

6. Line 123 – 124. The description of the RI prediction is not sufficient. A little more background should be provided. It appears molecular formulas are needed as input for the prediction. More details of the input are needed. Also, how the predictions will be used in this study were not mentioned.

7. Line 150. In prior description (Line 122), the authors claimed that the uncertainty for RI values are 0.02 – 0.03. Here, the authors argue that change by 0.02 – 0.03 is a "substantial enhance effect" between the two temperatures, despite this change is

on par with the uncertainty. A better case needs to be demonstrated regarding the enhancement of RI values at low temperature. In particular, this argument is a main result in this work.

8. Section 3.3. and Figure 2. From the chemical composition measured by ESI-TOFMS, the authors claimed that the largely different mass spectra are observed under lower temperature, due to enhanced oligomerization. It is unclear, however, how the changed temperature affected the chemical mechanisms that lead to different products. The discussion of higher oligomer formation under low temperature might be reasonable. But the difference in the monomer range does not make sense. The monomer products should be from the various RO2 chemistry regardless of the temperature. They should follow the same pattern. But in Figure 2, the monomer range in the two mass spectra show very different results. I wonder if the mass spectra shown here are reproducible? Two experiments were carried out for each condition. Do they show consistent mass spectra? Clark et al (2016, 50, 5564, Environ. Sci. Technol.) showed isoprene SOA ESI-TOFMS mass spectra under different temperatures. Similar to this work here, very different results were observed in the oligomer range, but not necessarily for monomers. Better clear discussions are need here, rather than speculations.

9. Section 3.4. It appears to me that the validation of the authors' argument using calculation method is important to extend further. Since it is a short manuscript, I suggest the authors include the results based on the prediction in the main text.

Technical comments: 1. Line 29. Change "heavy" to "heavily".

2. Line 62 – 68. When referring to prior temperature effect studies, it would be helpful to mention at what temperatures were those results observed.

3. In Figure 4, the y-axis between a and c, as well as b and d, are different. I suggest the authors use the same y-aixs ranges for easier comparison.

---

## Author Comment (AC1) · 7 May 2020

**Response to the comments of Reviewer #1**

*Li et al. presented a study that examined the effects of temperature on the optical properties and chemical composition of secondary organic aerosols formed from the OH photooxidation of n-dodecane. The authors found that oligomers were formed at low temperatures, and these oligomers resulted in higher RI values being measured. This paper is potentially useful to the SOA community. However, there are some important issues that the authors need to address before the manuscript can be considered for publication.*

Response: We thank Anonymous Referee #1 for the review and the positive evaluation of our manuscript. We have fully considered the comments and responded to these comments below in blue text. The revisions in the manuscript are highlighted in yellow color. The response and changes are listed below.

*Major comments:*

*1. Why were experiments conducted under dry conditions?*

Optical properties of the SOA can be affected by many factors, in order to study the temperature affect, other factors must be kept unchanged, so the humidity of the experiments must be constant and cannot be changed. The experiments were conducted under dry conditions (RH < 5%). Because when the temperature changes, the saturated vapor pressure of water changes, that is to say, if the RH is consistent at different temperatures, the concentration of the water is not consistent; when the concentration of water is the same, the RH is different. So choosing other humidity (non-dry conditions) will introduce new problems, we can only choose dry conditions. And we have added the related statement in the main text (Page 3, Line 88-93).

*2. Table 1 showed used of 43 ppb at low temperature vs. 58 ppb at high temperature. Why wasn't the same amount of n-dodecane used?*

Actually, the concentration of *n*-dodecane is tested with PTR-QMS, and the calibration of the PTR-QMS's response to *n*-dodecane is achieved through permeation tubes. According to our experimental design, the expected concentration of *n*-dodecane is 50 ppb, which is introducing 2 μL liquid *n*-dodecane into 5 m$^3$ smog chamber. As the injection volume of *n*-dodecane is 2 μL, volume error during injection is inevitable, which will influence the concentration of *n*-dodecane in the chamber. Nevertheless, the relative small differences in *n*-dodecane concentration (43-50 ppb at low temperature and 52-58 ppb at high temperature) likely have little influence in SOA composition and optical properties. And we have added the related statement in the main text (Page 4, Line 100-105).

*3. Do the authors know how the use of different temperatures will affect the loss rates of particles to the chamber walls? Is there a possibility that the observations of the SOA mass, composition and optical properties made by the authors can be explained partly by differences of particle wall loss rates at different temperatures?*

We have measured the wall loss rates of particles under both room and low temperatures, and found that the wall loss rate under low temperature condition (0.0025 ~ 0.0028 min$^{-1}$) is larger than that under room temperature condition (0.0018 ~ 0.0020 min$^{-1}$). However, this difference in particle wall loss rate can only slightly change the SOA mass concentration, but not the particle chemical composition. Therefore, it is unlikely to change the optical properties.

Nevertheless, the difference in vapor wall loss rates may change the particle composition and optical properties. The low temperature can enhance the loss rates of higher-volatility compounds (while for lower-volatility compounds, their dominant fates are condensation so temperature may affect little on their losses), which may lead to their lower proportions in SOA particles. As their RIs are generally lower than lower-volatility compounds (Li et al., 2018), this proportion change can probably enhance SOA RI at low temperature. Therefore, the difference in wall losses and gas-particle partitioning of gas-phase products might partially contribute to the RI enhancement

under low temperature condition.

We have added the above contents in the revised paper (Page 10, Line 297-308).

Reference:

Li, K., Li, J., Wang, W., Li, J., Peng, C., Wang, D., and Ge, M.: Effects of gas-particle partitioning on refractive index and chemical composition of m-xylene secondary organic aerosol, J. Phys. Chem. A, 122, 12, 3250-3260, 10.1021/acs.jpca.7b12792, 2018.

*4. In page 5 line 144, the authors state that "During this period, the optical properties of the particles tend to be stable and will not change much." This sentence is ambiguous and needs to be clarified. What optical property is the authors referring to? RI value? Or are they referring to the absorption spectra?*

The optical property here refers to the extinction coefficients ($\alpha_{ext}$) of the particles. During the last 1 h of the experiments, the extinction coefficients ($\alpha_{ext}$) measured by the CRDS at 532 nm and PAX at 375 nm tend to be stable; at the same time, the surface mean diameter (D) of the particles tend to be stable and does not change much. When the D and $\alpha_{ext}$ are stable, the extinction efficiency ($Q_{ext}$) will be stable (as shown in the following equation). A fixed set of D and $Q_{ext}$ results in a fixed RI value.

$$Q_{ext} = \frac{4\alpha_{ext}}{N\pi D^2}$$

We have rephrased this sentence to "During this period, the surface mean diameter and the extinction coefficients ($\alpha_{ext}$) of the particles tended to be stable and will not change much" (Page 7, Line 210-211)

*5. Explain the rationale behind tracking the absorption at 532 and 375 nm.*

The CRDS measures the total optical extinction coefficient ($\alpha_{ext}$) of the SOA, and $\alpha_{ext} = \alpha_{abs}$ (absorption coefficient) $+ \alpha_{sca}$ (scattering coefficient). In order to calculate the real (n) and imaginary (i) part of RI values, both the $\alpha_{abs}$ and the $\alpha_{sca}$ need to be known. So the absorption at 532 nm is measured with a UV-Vis light spectrometer (Avantes 2048F). With the measured $\alpha_{ext}$ and $\alpha_{abs}$, the $\alpha_{sca}$ will be calculated.

For the absorption at 375 nm, the rationale is as following: The photoacoustic extinctiometer (PAX-375, Droplet Measurement Technologies) directly measures in-situ light absorption and scattering of aerosol particles at 375 nm, from which the $\alpha_{abs}$ and $\alpha_{sca}$ can be derived.

*6. The authors did not use seed aerosols in this study to promote gas-to-aerosol partitioning. Hence, I expect substantial vapor wall loss in these experiments, and the extent of vapor wall loss is likely to be different at 5 ℃ vs. 25 ℃. Is it possible that the authors are not detecting some products (due to their loss to the chamber walls) that can contribute to SOA optical properties?*

We agree that the gas-particle partitioning can influence the particle composition and optical properties. As shown in our previous study (Li et al., 2017), the presence of seeds can promote the condensation of low-molecular-weight products and decrease the real part of the RI of *n*-dodecane SOA under low $NO_X$ conditions. The experimental conditions in that study are very similar to those of room temperature condition in this study. Therefore, we can expect that similar results can be found in this study at room temperature. However, even if we perform the experiments with seeds in this study, the gas-particle partitioning can also be different at different temperature. In other words, we cannot rule out the contribution of different gas-particle partitioning at different temperatures.

As we have discussed in our response to comment #3, the difference in gas-particle partitioning of products might partially contribute to the RI enhancement under low temperature condition.

Reference:

Li, J., Li, K., Wang, W., Wang, J., Peng, C., and Ge, M.: Optical properties of secondary organic aerosols derived from long-chain alkanes under various $NO_x$ and seed conditions, Sci. Total Environ., 579, 1699-1705, 10.1016/j.scitotenv.2016.11.189, 2017.

*7. In section 3.5, the authors tried to relate their results to observations made during*

*winter haze episodes in China. I advise the authors to be more circumspect in the extrapolation of their results to ambient observations since NOx concentrations are likely substantial during winter haze episodes in China. The authors performed a study under low-NOx conditions. Under high NOx conditions, I expect the reaction mechanism of n-dodecane OH photooxidation to be different. For example, more fragmentation will likely happen, which will result in the formation of more volatile products. If this is the case for both 5 ℃ and 25 ℃ conditions, there may not be significant differences between SOA composition and their RI values under high NOx conditions, which would imply that temperature does not play a big role in the DRF of SOA formed under areas with significant NOx concentrations, like China.*

We agree that the reaction mechanism of *n*-dodecane under high $NO_X$ conditions is different from that under low $NO_X$ conditions. The temperature effects in the presence of $NO_X$ are indeed very important and will be investigated in future studies. Therefore, we have revised this part in the manuscript "The enhancement in light extinction of SOA and oligomer compositions formed under low temperature condition might provide some possible inspiration for the regional visibility issues, especially the suburban areas. It had been reported that UV-scattering particles in the boundary layer could accelerate photochemical reactions and haze production (Sun et al., 2014). The observations above showed that the scattering property of formed SOA increased under low temperature condition, which might provide one possible reason for the rapid occurrence of haze in suburban areas in winter. According to field observations, haze occurred frequently in winter, especially in China (Cheng et al., 2016; Huang et al., 2014; Guo et al., 2014; Parrish et al., 2007). When haze occurred, it was often accompanied by high $NO_X$, especially in urban areas. The temperature effects under high-$NO_X$ conditions are also important and need to be investigated in future studies." (Page 11, Line 317-324)

*Minor comments:*

*8. Inconsistent tenses. The authors switch between using past and present tenses in some parts of the manuscript. Please correct this.*

Thank you for the helpful comments. We have corrected this in the manuscript.

Page 7, Line 209: are → were; reaches → reached;

Page 7, Line 210: are → were;

Page 7, Line 211: tend → tended;

Page 7, Line 215: are → were;

Page 7, Line 216: have → had;

Page 8, Line 223: indicate → indicated;

Page 8, Line 249: are → were;

Page 9, Line 257: changes → changed.

---

## Author Comment (AC2) · 7 May 2020

**Response to the comments of Reviewer #2**

*This work by Li et al. describes measurements of optical properties in relation with chemical composition of n-dodecane SOA under low-NO$_x$ and two different temperatures (5 ℃ and 25 ℃). The authors found that under low temperature, the real part of the refractive index (RI) at 375 nm and 532 nm is enhanced, corresponding to substantial oligomer formation. The authors hence conclude that the enhanced oligomer formation under low temperature lead to the higher RI. The results could be relevant to low visibility in urban areas during wintertime. Overall, the manuscript is well written and demonstrates new findings regarding SOA optical properties vs. chemical compositions. But a few major concerns need to be addressed before this manuscript can be considered publishing.*

Response: We thank Anonymous Referee #2 for the review and the positive evaluation of our manuscript. We have fully considered the comments and responded to these comments below in blue text. The revisions in the manuscript are highlighted in yellow color. The response and changes are listed below.

*Specific comments:*

*1. Line 49 – 52. Although it is true that few studies have examined temperature effects on RI values of SOA, temperature effect studies on SOA formation and composition are not limited. There are many other studies on SOA formation and temperature effects not cited. For example, Warren et al., 2009, 43, 3548, Atmos. Environ.; Emanuelsson et al., 2013, 117, 10346, J. Phys. Chem. A.; Price et al., 2016, 50, 1216, Aerosol. Sci. Technol.; Boyd et al., 2017, 51, 7831, Environ. Sci. Technol.; Zhao et al., 2019, 3, 2549, ACS Earth Space Chem., etc.*

We have revised this part in the manuscript, and the references mentioned above have been cited:

"There are also many studies investigating temperature effects on SOA formation and composition (Takekawa et al., 2003; Svendby et al., 2008; Clark et al., 2016;

Lamkaddam et al., 2016; Huang et al., 2017; Qing Mu and Gerhard Lammel, 2018; Zhao et al., 2019; Boyd et al., 2017; Price et al., 2016; Emanuelsson et al., 2013; Qi et al., 2010; Warren et al., 2009); however, works on the effect of temperature on the SOA RI are limited (Kim et al., 2014)." (Page 2, Line 52-56)

References:

Boyd, C. M., Nah, T., Xu, L., Berkemeier, T., and Ng, N. L.: Secondary organic aerosol (SOA) from nitrate radical oxidation of monoterpenes: Effects of temperature, dilution, and humidity on aerosol formation, mixing, and evaporation, Environ. Sci. Technol., 51, 7831-7841, 10.1021/acs.est.7b01460, 2017.

Emanuelsson, E. U., Watne, A. K., Lutz, A., Ljungstrom, E., and Hallquist, M.: Influence of humidity, temperature, and radicals on the formation and thermal properties of secondary organic aerosol (SOA) from ozonolysis of beta-pinene, J. Phys. Chem. A, 117, 10346-10358, 10.1021/jp4010218, 2013.

Huang, W., Saathoff, H., Pajunoja, A., Shen, X., Naumann, K.-H., Wagner, R., Virtanen, A., Leisner, T., and Mohr, C.: α-Pinene secondary organic aerosol at low temperature: Chemical composition and implications for particle viscosity, Atmos. Chem. Phys., 18, 2883-2898, https://doi.org/10.5194/acp-18-2883-2018, 2018.

Kim, H., Liu, S., Russell, L. M., and Paulson, S. E.: Dependence of real refractive indices on O:C, H:C and mass fragments of secondary organic aerosol generated from ozonolysis and photooxidation of limonene and alpha-pinene, Aerosol Sci. Technol., 48, 498-507, 10.1080/02786826.2014.893278, 2014.

Lamkaddam, H., Gratien, A., Pangui, E., Cazaunau, M., Picquet-Varrault, B., and Doussin, J.-F.: High-NOx photooxidation of n-dodecane: Temperature dependence of SOA formation, Environ. Sci. Technol., 10.1021/acs.est.6b03821, 2016.

Price, D. J., Kacarab, M., Cocker, D. R., Purvis-Roberts, K. L., and Silva, P. J.: Effects of temperature on the formation of secondary organic aerosol from amine precursors, Aerosol Sci. Technol., 50, 1216-1226, 10.1080/02786826.2016.1236182, 2016.

Qi, L., Nakao, S., Tang, P., and R., C. I. D.: Temperature effect on physical and chemical properties of secondary organic aerosol from m-xylene photooxidation, Atmos. Chem. Phys., 10, 3847-3854, 2010.

Qing Mu, M. S., Mega Octaviani, Nan Ma, Aijun Ding, Hang Su,, and Gerhard Lammel, U. P., Yafang Cheng: Temperature effect on phase state and reactivity controls atmospheric multiphase chemistry and transport of PAHs, Sci. Adv., 4, 10.1126/sciadv.aap7314, 2018.

Svendby, T. M., Lazaridis, M., and Tørseth, K.: Temperature dependent secondary organic aerosol formation from terpenes and aromatics, J. Atmos. Chem., 59, 25-46, 10.1007/s10874-007-9093-7, 2008.

Takekawa, H., Minoura, H., and Yamazaki, S.: Temperature dependence of secondary organic aerosol formation by photo-oxidation of hydrocarbons, Atmos. Environ., 37, 3413-3424, 10.1016/s1352-2310(03)00359-5, 2003.

Warren, B., Austin, R. L., and Cocker, D. R.: Temperature dependence of secondary organic aerosol,

Atmos. Environ., 43, 3548-3555, 10.1016/j.atmosenv.2009.04.011, 2009.

Zhao, Z., Le, C., Xu, Q., Peng, W., Jiang, H., Lin, Y.-H., Cocker, D. R., and Zhang, H.: Compositional evolution of secondary organic aerosol as temperature and relative humidity cycle in atmospherically relevant ranges, ACS Earth Space Chem., 3, 2549-2558, 10.1021/acsearthspacechem.9b00232, 2019.

*2. Line 78. Although details of the chamber can be referred to previous work, important and fundamental characteristics of the chamber still needs to be provided. For example, the material and volume of the chamber.*

We have added the important and fundamental characteristics of the chamber in the **2.1 Smog Chamber Experiments** part:

"The chamber consisted of two 5 $m^3$ reactors made of fluorinated ethylene propylene (FEP) Teflon-film, which were housed in a thermally isolated enclosure. The temperature in the chamber was accurately controlled by high-power air conditioner in the range of -10 - 40 °C." (Page 3, Line 82-84)

*3. Some important details of the chamber experiments are missing. Why were instruments not connected before temperature was stabilized? SOA mass concentration in each experiment should be reported (in Table 1 or in the main text). How long were filter samples collected and at what flow rate? After the filters were dissolved in filter, were the solutions sonicated? If not, how good were the extraction recoveries?*

Actually, the instruments were connected at room temperature condition, when the temperature dropped to 5 ℃ and stabilized, the data measured by the instruments would to counted as valid data. And we have added this explanation in the main text (Page 3, Line 97-99).

We have added SOA mass concentration in Table 1 (Page 17, Line 605):

**Table 1. The initial conditions of the smog-chamber experiments.**

| Experiments No.[a] | HC (ppb) | H₂O₂ (ppm) | NOₓ (ppb) | RH (%) | Temperature (℃) | Mass[b] (μg/m³) |
|---|---|---|---|---|---|---|
| Dod-R-1 | 58 | 1.03 | <1 | <5 | 25 | 155 |
| Dod-R-2 | 52 | 1.03 | <1 | <5 | 25 | 135 |

| Dod-L-1 | 43 | 1.07 | <1 | <5 | 5 | 128 |
| Dod-L-2 | 50 | 1.09 | <1 | <5 | 5 | 133 |

Each filter sample was collected for 30 min at 10 L/min flow rate. Then the filters were put into 5 mL methanol (99.9%, Fisher Chemical) and sonicated for 30 min. We have added this part in the manuscript (Page 4, Line 119-120).

*4. How did the extinction coefficient and extinction efficiency evolve over the course of an experiment?*

With the progress of the reaction, the surface mean diameter of the particles gradually increased, and the extinction coefficient and extinction efficiency of the particles increased as well. We chose two sets of experiments under two temperature conditions as examples. The figure below has been added in the Supporting Information of the manuscript. (Supporting Information, Page 7, Figure S4.)

[Figure]

Figure S4. Evolution of optical parameters at 532 nm. Extinction coefficient (a) and efficiency (b) at room temperature; extinction coefficient (c) and efficiency (d) at low temperature.

*5. Line 122. It is unclear how was the RI value uncertainty estimated from the various uncertainties. An equation needs to be provided.*

We have added the estimating method and equations in the Supporting Information: (Supporting Information, Page 2, Line 28-43)

"The RI values are obtained with extinction efficiency ($Q_{ext}$) and surface mean

diameter ($D_s$). The uncertainty of the $D_s$ measured by SMPS is $\pm 1\%$. The uncertainty of the extinction efficiency is calculated as the following:

The extinction efficiency can be expressed as:

$$Q_{ext} = \alpha_{ext} / (\frac{1}{4} N\pi D^2)$$

The three variables $\alpha_{ext}$, N, D are independent and the uncertainty of them were 3%, 10% and 1%, respectively. Considering the propagation of uncertainty, we can obtain the variance formula:

$$\sigma_Q^2 = \sigma_\alpha^2 (\frac{\partial Q}{\partial \alpha})^2 + \sigma_N^2 (\frac{\partial Q}{\partial N})^2 + \sigma_D^2 (\frac{\partial Q}{\partial D})^2$$

Then the uncertainty of $Q_{ext}$ can be calculated as:

$$\frac{\sigma_Q}{Q} = \sqrt{\left(\frac{\sigma_\alpha}{\alpha}\right)^2 + \left(\frac{\sigma_N}{N}\right)^2 + 4(\frac{\sigma_D}{D})^2} = \sqrt{0.03^2 + 0.1^2 + 4 \times 0.01^2}$$

$$= \sqrt{0.0113} = 10.6\%$$

The measured extinction efficiency ($Q_{ext,mea}$) is compared to calculated extinction efficiency ($Q_{ext,cal}$). The best-fit RI value is determined by minimizing the following reduced merit function ($\chi_r$):

$$\chi_r = \frac{1}{N} \sum_{i=1}^{N} (Q_{ext,mea} - Q_{ext,cal}(n, k))_i^2$$

The uncertainty of the retrieval method is $\pm 0.002$, and the uncertainty of the measured extinction efficiency is $\pm 10.6\%$, resulting in the final uncertainty of the retrieved RI value to be about 0.02-0.03."

*6. Line 123 – 124. The description of the RI prediction is not sufficient. A little more background should be provided. It appears molecular formulas are needed as input for the prediction. More details of the input are needed. Also, how the predictions will be used in this study were not mentioned.*

We have added the following section to the main text (Page 6, Line 166-182):

"**2.3.2 Calculation Method of the QSPR Based on the Molecular Formula**

Using the molecular formula obtained from ESI-TOF-MS, we calculated the RI

values of the products in SOA with the quantitative structure–property relationship (QSPR) method (Redmond and Thompson, 2011). The QSPR can be expressed with Equation (9):

$$RI_{predicted} = 0.031717(\mu) + 0.0006087(\alpha) - 3.0227\left(\frac{\rho_m}{M}\right) + 1.38709 \quad (9)$$

where $\mu$ is the unsaturation of the molecular formula, $\alpha$ is the polarizability of the molecular formula, $\rho_m$ is the mass density (g/cm³), and $M$ is the molar mass (g/mol).

The mass density of the compound is estimated by AIM model, the detailes of which can be referred to Girolami (1994).

$\mu$ is calculated through the conventional approach, which is used in many organic chemistry texts, Equation (10)

$$\mu = (\#C + 1) - 0.5(\#H - \#N) \quad (10)$$

where #C,#H, and #N are the number of the C, H, and N respectively.

$\alpha$ is calculated based on the molecular formula of the compound, it can be expressed by Equation (11):

$$\alpha = 1.51(\#C) + 0.17(\#H) + 0.57(\#O) + 1.05(\#N) + 2.99(\#S) + 2.48(\#P) +$$
$$0.22(\#F) + 2.16(\#Cl) + 3.29(\#Br) + 5.45(\#I) + 0.32 \quad (11)$$

where # is the number of the atoms of each element in the molecular formula.

The calculated RI values of products were used to link chemical composition and optical properties, and to explain the obseved RI differences at different temperatures in Sect. 3.4."

Reference:

Redmond, H., and Thompson, J. E.: Evaluation of a quantitative structure-property relationship (QSPR) for predicting mid-visible refractive index of secondary organic aerosol (SOA), Phys. Chem. Chem. Phys. : PCCP, 13, 6872-6882, 10.1039/c0cp02270e, 2011.
Girolami, G. S.: A simple" back of the envelope" method for estimating the densities and molecular volumes of liquids and solids, J. Chem. Edu., 71, 11, 962, 1994.

*7. Line 150. In prior description (Line 122), the authors claimed that the uncertainty for RI values are 0.02 − 0.03. Here, the authors argue that change by 0.02 − 0.03 is a "substantial enhance effect" between the two temperatures, despite this change is on par with the uncertainty. A better case needs to be demonstrated regarding the enhancement of RI values at low temperature. In particular, this argument is a main result in this work.*

In order to more intuitively display the RI variation tendency of SOA generated under two temperature conditions, we have added Figure 2 in the main text. As shown in Figure 2, the RI under low temperature conditions is enhanced whether at 532 nm or 375 nm. The enhancement of RI values under low temperature condition is still valid with experiment uncertainty. We have changed "a substantial enhance effect" into "an enhance effect" in the revised manuscript. (Page 8, Line 219)

[Figure]

Figure 2. Variation tendency of the RI values as a function of surface mean diameter for SOA produced under room and low temperature conditions at (a) 532 nm and (b) 375 nm.

*8. Section 3.3. and Figure 2. From the chemical composition measured by ESITOFMS, the authors claimed that the largely different mass spectra are observed under lower temperature, due to enhanced oligomerization. It is unclear, however, how the changed temperature affected the chemical mechanisms that lead to different products. The discussion of higher oligomer formation under low temperature might be reasonable. But the difference in the monomer range does not make sense. The monomer products should be from the various $RO_2$ chemistry regardless of the temperature. They should*

*follow the same pattern. But in Figure 2, the monomer range in the two mass spectra show very different results. I wonder if the mass spectra shown here are reproducible? Two experiments were carried out for each condition. Do they show consistent mass spectra? Clark et al (2016, 50, 5564, Environ. Sci. Technol.) showed isoprene SOA ESI-TOFMS mass spectra under different temperatures. Similar to this work here, very different results were observed in the oligomer range, but not necessarily for monomers. Better clear discussions are need here, rather than speculations.*

We have updated Figure 2b (now is Figure 3b) by removing impurity interferences at ~200 and 300 amu:

[Figure]

Figure 3. Mass spectra of *n*-dodecane SOA obtained by ESI-TOF-MS in positive ion mode. (a) low temperature condition; (b) room temperature condition.

The mass spectra under the same temperature condition are reproducible, see below the mass spectra of two experiments under room temperature. Though there are some slight differences in peak intensities, these two mass spectra are generally similar.

[Figure]

As the reviewer said, Clark et al. (2016) found that the SOA formed from isoprene had similar mass spectra at the monomer range at different temperatures. However, the precursors are very different (isoprene vs *n*-dodecane), which have very different oxidation pathways and partitioning processes. Hence, it is possible that we observe different results in mass spectra at monomer range.

As we have mentioned in the manuscript, the different oxidation degree and partitioning process have made the difference in monomer range. At room temperature, more OH oxidation steps in the gas phase can lead to the formation of some fragmentation products that may not be observed in the low temperature. For example, we have observed the high intensity of M/Z 195 $C_{10}H_{20}O_2$ (Relative Intensity 0.39), M/Z 211 $C_{10}H_{20}O_3$ (Relative Intensity 0.13) at room temperature but very low intensity (0.012, 0.041, respectively) at low temperature, and M/Z 195 and 211 are likely products from gas-phase OH oxidation. This process may be different from isoprene (Clark et al., 2016), because for *n*-dodecane the carbon number is high and volatility is low and fewer oxidation steps are needed before partitioning, while for isoprene there is only 5 carbon atoms so more oxidation steps are needed before partitioning no matter at room or low temperature.

Overall, the differences in monomer range still make sense.

We have added related statement in the main text. (Page 9, Line 262-273)

Reference:

Clark, C. H., Kacarab, M., Nakao, S., Asa-Awuku, A., Sato, K., and Cocker, D. R., 3rd: Temperature effects on secondary organic aerosol (SOA) from the dark ozonolysis and photo-oxidation of isoprene, Environmental Science & Technology, 10.1021/acs.est.5b05524, 2016.

*9. Section 3.4. It appears to me that the validation of the authors' argument using calculation method is important to extend further. Since it is a short manuscript, I suggest the authors include the results based on the prediction in the main text.*

Thanks for this suggestion. We have moved the calculation method to the main text, and they are highlighted in yellow in the "2.3 Calculation Method of RI Values" Section. (Page 4-6, Line 125-189). Figure S3, Figure S5, and Table S2 are also moved to the main text, and they have updated as Figure 4, Figure 6, and Table 2.

*Technical comments:*

1. *Line 29. Change "heavy" to "heavily".*

Page 1, Line 32: This has been corrected.

2. *Line 62 – 68. When referring to prior temperature effect studies, it would be helpful to mention at what temperatures were those results observed.*

Page 2, Line 67: The temperature ranges have been added.

3. *In Figure 4, the y-axis between a and c, as well as b and d, are different. I suggest the authors use the same y-aixs ranges for easier comparison.*

Thanks for this suggestion. This has been corrected, and Figure 4 has been updated as Figure 7 in the main text.

---

## Author Response (AR2)

**Response to the comments of Reviewer #2**

*The revised manuscript has carefully addressed my earlier comments. The paper quality has clearly improved. I have two additional comments regarding the revised manuscript:*

Response: We thank Anonymous Referee #2 for the review and the positive evaluation of our manuscript. We have fully considered the comments and responded to these comments below in blue text. The revisions in the manuscript are highlighted in yellow color. The response and changes are listed below.

*1. It is interesting that there was not enhancement in SOA mass under lower temperature. This is somewhat in contract with prior research. Some discussion is needed to explain the observation. It appears that under both conditions, the SOA mass reached maximum before the end of the experiments. So I do not think the reactivity is a major issue here. The authors should consider a few other aspects (i.e., chemistry, wall loss, etc.) in detailed discussion.*

Thank you for the valuable comments and suggestions. The particle wall loss rates under both room and low temperature conditions had been measured. The wall loss rate under low temperature condition ($0.0025 \sim 0.0028$ min$^{-1}$) was larger than that under room temperature condition ($0.0018 \sim 0.0020$ min$^{-1}$). After wall loss correction, the SOA mass under low temperature condition was higher than that under room temperature condition. We have added the related statement in the main text and Table 1 (Page 10, Line 306-309; Page 17, Line 611-615)

*2. In these experiments, after SOA mass reached maximum, there are 2-3 hours remaining but why did sample collection only last for 30 min? Longer collection could allow for better accuracy in the optical and compositional measurements.*

The optical properties of the formed particles were analyzed after the mass

concentration of the aerosol reached the maximum. During the following one to two hours, the surface mean diameter and the extinction coefficients of the particles tended to be stable and would not change much. Then, we collected the particles on the film to analyze its chemical composition in the same period. The sample collection time was chosen to make sure the signal of the collected filter is much higher than the background of the blank filter in mass spectrometry. Another principle is less sample volume used in this process. If the membrane extraction time is too long, the chamber volume will decrease too much. So, a sampling time of 30 minutes is chosen. Thanks for the suggestion, and we would use longer collection time when the low mass concentration of particles are encountered in our future research work. We have added the related statement in the main text. (Page 4, Line 114-121)